# Updated Trends on the Biodiscovery of New Marine Natural Products from Invertebrates

**DOI:** 10.3390/md20060389

**Published:** 2022-06-09

**Authors:** Ricardo Calado, Renato Mamede, Sónia Cruz, Miguel C. Leal

**Affiliations:** ECOMARE, CESAM-Centre for Environmental and Marine Studies, Department of Biology, University of Aveiro, Santiago University Campus, 3810-193 Aveiro, Portugal; renatomamede@ua.pt (R.M.); sonia.cruz@ua.pt (S.C.)

**Keywords:** biomolecules, bioprospecting, corals, sponges, zoogeography

## Abstract

From 1990–2019, a total of 15,442 New Marine Natural Products from Invertebrates (NMNPIs) were reported. The 2010s saw the most prolific decade of biodiscovery, with 5630 NMNPIs recorded. The phyla that contributed most biomolecules were the Porifera (sponges) (47.2%, 2659 NMNPIs) and the Cnidaria (35.3%, 1989 NMNPIs). The prevalence of these two phyla as the main sources of NMNPIs became more pronounced in the 2010s. The tropical areas of the Pacific Ocean yielded more NMNPIs, most likely due to the remarkable biodiversity of coral reefs. The Indo-Burma biodiversity hotspot (BH) was the most relevant area for the biodiscovery of NMNPIs in the 2010s, accounting for nearly one-third (1819 NMNPIs) of the total and surpassing the top BH from the 1990s and the 2000s (the Sea of Japan and the Caribbean Islands, respectively). The Chinese exclusive economic zone (EEZ) alone contributed nearly one-quarter (24.7%) of all NMNPIs recorded during the 2010s, displacing Japan’s leading role from the 1990s and the 2000s. With the biodiscovery of these biomolecules steadily decreasing since 2012, it is uncertain whether this decline has been caused by lower bioprospecting efforts or the potential exhaustion of chemodiversity from traditional marine invertebrate sources.

## 1. Introduction

The marine realm is likely the largest reservoir of untapped chemical diversity on the planet. Harboring most of the branches of the tree of life, oceans feature a remarkable biodiversity distributed along contrasting habitats. From the warm shallow waters of tropical coral reefs to the cold waters of the poles and the dark depths of the ocean floor where no sunlight can penetrate, marine life has managed to adapt and thrive in forms and shapes beyond our imagination [1].

While thousands of New Marine Natural Products from Invertebrates (NMNPIs) have been reported to date [2], it is likely that the chemical diversity of these organisms is yet to be fully untapped. Each year, more and more NMNPIs are reported, with some of these molecules showcasing an array of bioactivities that make them highly appealing for high-end products and uses [2]. From pharmaceutical and biomedical applications to cosmeceutical and nutraceutical solutions, the marketing of NMNPIs is a thriving sector [3], powering a new bioeconomy paradigm [4].

Surveying the chemical diversity of world oceans requires the use of optimized and efficient sampling techniques, while simultaneously accounting for several legal and ethical issues. These issues are covered by international treaties, such as the Convention on Biological Diversity and the Nagoya Protocol, that provide a framework for fair and equitable benefit sharing from the use of biological resources [5]. The biodiscovery of NMNPIs is an excellent case study for understanding how bio-based activity can contribute to the capacity building of developing countries, which may hold high levels of marine biodiversity but insufficient technical and financial resources to capitalize on sustainable bio-based activities [6].

Bioprospecting for NMNPIs has often neglected accurate taxonomic information records, as well as the geographic origins of source biological material [7]. The identification of biodiversity hotspots and species distribution, along with potential latitudinal and/or longitudinal gradients, might allow researchers to forecast the distribution of NMNPIs in the ocean [8]. However, the absence of accurate taxonomic and biogeographic information has hindered researchers from performing cross-disciplinary meta-analyses to maximize the success of present and future efforts on the ongoing quest for NMNPIs.

Overall, by identifying the taxonomic and biogeographic trends associated with the biodiscovery of NMNPIs, it is possible to provide relevant insights on how this important driver of future blue biotechnological solutions will likely evolve. The present work builds upon a previous study developed by Leal et al. [9], which pinpointed marine invertebrate taxa that yielded more NMNPIs from 1990 to 2009 and their geographic origin, and how the trends of these two features have evolved over time. Updated trends for the worldwide biodiscovery of NMNPIs over the last three decades (1990–2019) are now presented and critically discussed to update the overarching view of marine bioprospecting trends, both taxonomically and geographically.

## 2. Results

### 2.1. Taxonomical Trends

From 1990 to 2019, eleven different phyla (Porifera, Cnidaria, Echinodermata, Chordata, Mollusca, Annelida, Bryozoa, Hemichordata, Arthropoda, Platyhelmintes, and Brachiopoda) yielded a total of 15,442 NMNPIs (Figure 1). During the period between 2010 and 2019 alone, 5630 NMNPIs were reported, with none originating from phyla Platyhelmintes or Brachiopoda (unlike in the 1990s and the 2000s). As in previous decades, the Porifera contributed the highest percentage of NMNPIs during the period of 2010–2019 with 47.2% (2659 NMNPIs), followed by the Cnidaria with 35.3% (1989 NMNPIs). The number of NMNPIs from the Porifera increased 6.5% over the periods ranging from 2000–2009 and 2010–2019 (from 2496 to 2659 NMNPIs), while for those originating from the Cnidaria, the increase recorded was of 12.2% between the same periods (from 1773 to 1989 NMNPIs) (Figure 1). It is worth highlighting that after 2012, the number of NMNPIs annually reported has steadily trended downward, with the lowest value reported from the 2010s being 413 NMNPIs in 2019 (Appendix A).

#### 2.1.1. Porifera

Class Demospongiae accounted for more than 90% of all NMNPIs recorded from the Porifera from 2010–2019, with more than half of all NMNPIs (55%) in this taxon being yielded from 10 sponge families alone: Thorectidae, Petrosiidae, Dysideidae, Plakinidae, Ancorinidae, Halichondriidae, Spongiidae, Theonellidae, Chalinidae, and Agelasidae. Family Thorectidae accounted for 11% of all NMNPIs reported for this phylum (at 295 NMNPIs) and family Petrosiidae accounted for 9% (at 239 NMNPIs) (Figure 2); these numbers represented an increase of 13% and 92%, respectively, when compared to the number of NMNPIs reported from these taxa during the period of 2000–2009. The sponge genera that contributed more NMNPIs during this period were *Xestospongia*, *Agelas* and *Plakortis* (with 132, 118 and 113 NMNPIs, respectively). However, nearly one-third of these NMNPIs were not identified to species level. While the need for accurate species identification for NMNPIs has already been advocated, with some scientific journals having well-established guidelines on this topic [7], taxonomic issues are often still over-looked when reporting NMNPIs. The identification of sponges at species level has long been recognized as challenging [10]; however, tools that have already been available for some years (e.g., World Porifera Database, https://www.marinespecies.org/porifera/ (accessed on 23 April 2022)) can help researchers solve this bottleneck [11]. Two sponge species must be highlighted due to the remarkable number of NMNPIs yielded from them during the 2010s: the tube sponge *Theonella swinhoei* (with 75 NMNPIs) and the giant barrel sponge *Xestospongia testudinaria* (with 74 NMNPIs). Further details on NMNPIs from the Porifera and their associated trends over the last decades are presented in Table 1.

#### 2.1.2. Cnidaria

Concerning the Cnidaria, 95% of all NMNPIs reported from 2010–2019 originated from order Alcyonacea. Indeed, nearly one out of five NMNPIs reported during 2010–2019 were screened from a species within family Alcyoniidae. Families Ellisellidae, Gorgoniidae, Nephtheidae, and Plexauridae jointly contributed another 30% (593 NMNPIs) of all NMNPIs from this phylum (Figure 3). The contribution of NMNPIs from family Alcyoniidae more than doubled when compared to data reported from 2000–2009 and more than tripled in comparison to the data from 1990–1999. The three genera of alcyoniid corals that yielded more NMNPIs in the 2010s were *Sinularia*, *Sarcophyton*, and *Lobophyton* (with 393, 251 and 146 NMNPIs, respectively) (Figure 4), representing nearly 15% of all new biomolecules reported during this decade. The three single cnidarian species yielding more NMNPIs during the 2010s were the asparagus sea fan *Dichotella gemmacea* (with 82 NMNPIs), the leather coral *Sarcophyton trocheliophorum* (with 69 NMNPIs), and the sea whip *Junceella fragilis* (with 55 NMNPIs). A total of 47 NMNPIs were also identified from the species *Sinularia flexibilis* during this period. However, one may question whether this number may be even higher, as 87 NMNPIs yielded from this genus were retrieved from specimens that were not reported to species level (simply being termed as *Sinularia* sp.). It is certainly a challenge to accurately identify alcyoniid corals to species level, namely *Sinularia* [12], as hybridization can occur, and complexes of cryptic species continue to be identified [13]. As such, further studies must pursue the combination of both molecular and chemical fingerprinting [14]. Further details on the finding of NMNPs from the Cnidaria and their associated trends over the last decades are presented in Table 1.

#### 2.1.3. Other Phyla

Other than the Porifera and the Cnidaria, phyla Echinodermata, Chordata, and Mollusca were the ones that most contributed NMNPIs from 2010–2019 (379, 262, and 262 NMNPIs, respectively). Unlike the Porifera and the Cnidaria, which saw increases in the number of NMNPIs reported from 2010–2019 (when compared with the periods 1990–1999 and 2000–2009), the number of NMNPIs reported from phyla Echinodermata, Chordata, and Mollusca either decreased or remained unchanged during the period of 2010–2019. Further details on the findings of NMNP from phyla other than the Porifera and the Cnidaria, along with their associated trends over the last decades, are presented in Table 2.

#### 2.1.4. Bioprospecting Efforts and Natural Products Richness

The percentage of species yielding the total number of NMNPIs reported from 2010–2019 for the nine different phyla listed represents <1% of all species reported for these taxa in the WoRMS database [15]. The NMNPIs recorded from the Porifera between 2010 and 2019 were retrieved from 355 different species, which accounts for 3.9% of all species diversity reported for this phylum. It is worth highlighting that NMNPIs reported from orders Verongiida and Agelasida were generated by a number of species (19 and 16, respectively) representing more than one-fifth (20.7 and 22.9%, respectively) of all known extant sponges from these taxa. Order Poecilosclerida contributed the highest number of sponge species yielding NMNPIs from 2010–2019 (62 species), but this taxon is also one of the most speciose extant orders for sponge species (with these representing only 2.6% of all known species in this order). Concerning the Cnidaria, it is worth highlighting order Alcyonacea, as 5.1% of its >3400 species yielded NMNPIs from 2010–2019 (for a total of 178 soft coral species).

### 2.2. Geographical Trends

Despite the growing awareness of a need to provide adequate information on the place of origin of collected samples used to screen NMNPIs [7], it was still not possible to assign all compounds to a specific geographic region (see below).

#### 2.2.1. Latitudinal

From 2010–2019, more than 80% of the 5630 NMNPIs were recorded in the northern hemisphere, thus further enhancing the trend already reported from 1990–2009 (at ≈60% for this timeframe). Less than 1% of NMNPIs reported from 2010–2019 (35 NMNPIs) could not be attributed to one of the hemispheres, with this figure increasing to almost 3% when allocating NMNPIs to a polar, temperate, or tropical location (161 NMNPIs). About two-thirds of all NMNPIs reported from 2010–2019 originated from the tropics (3724 NMNPIs), with only 1% of NMNPIs originating from polar areas. While the aforementioned proportions of NMNPIs from the specimens collected in temperate vs. tropical regions in the southern hemisphere was 1:1.5, it was much more skewed towards the tropics in the northern hemisphere at 1:2.4. Concerning the trends from 2000–2009, the prevalence of NMNPIs sourced from tropical south-area specimens decreased by almost two-thirds (from 1731 to 629 NMNPIs), and nearly doubled in the tropical north (from 1533 to 3075 NMNPIs).

While until 2009 phylum Echinodermata contributed the highest number of species yielding NMNPIs within polar regions, between 2010 and 2019 this position was surpassed by phylum Bryozoa. Phylum Porifera dominated in temperate regions (with 910 NMNPIs) and yielded more than half of all NMNPIs reported in these locations from 2010–2019, with the Cnidaria only accounting for 15% of NMNPIs. Both phyla dominated in tropical areas, with the Cnidaria yielding 43.3% (1611 NMNPIs) and the Porifera yielding 45.4% (1692 NMNPIs). The cnidarian order Alcyonacea accounted for 13.1% (221 NMNPIs) and 41.9% (1561 NMNPIs) of all NMNPIs reported during the period of 2010–2019 from temperate and tropical regions, respectively. In tropical regions, the soft coral genera *Sinularia*, *Sarcophyton*, and *Lobophytum* yielded 625 NMNPIs alone, further highlighting the prevalence of the cnidarian family Alcyoniidae, which yielded one out of five NMNPIs reported from the tropics for the period from 2010 to 2019.

#### 2.2.2. Oceans and Continents

The Pacific Ocean, which played the leading role from 1990–2009 as the main source of species yielding NMNPIs, has continued this trend, accounting for 77% of all NMNPIs reported from 2010–2019. The Atlantic Ocean, which used to dominate over the Indian Ocean (at 19.6% over 12.8%, respectively), switched roles during the period of 2010–2019, with the Indian Ocean yielding slightly more NMNPIs than the Atlantic (10.7 vs. 9.9%). The Arctic and Antarctic Oceans combined accounted for less than 1.5% of all NMNPIs reported from 2010–2019. While the majority (40.8%) of NMNPIs recorded in the Antarctic were yielded from the Mollusca (20 NMNPIs which were mostly nudibranchs), the Porifera dominated in all other oceans, representing 38.1% of all NMNPIs in the Artic (8 NMNPIs), 44.6% of all NMNPIs in the Pacific (1943 NMNPIs), 53.2% of all NMNPIs in the Atlantic (295 NMNPIs), and 62.4% of all NMNPIs in the Indian Ocean (375 NMNPIs). It is worth highlighting that the Cnidaria nearly matched the Porifera in the Pacific, totaling 39.2% of all NMNPIs recorded in this ocean (1706 NMNPIs). The cnidarian order Alcyonacea (soft corals) was most the representative one in the Atlantic, Indian, and Pacific oceans, yielding 109, 110 and 1655 NMNPIs, respectively, for the period of 2010–2019. Concerning the genera of marine invertebrates that yielded more NMNPIs in the Atlantic, it is worth emphasizing the contributions of the sea fans *Pseudopterogorgia* (43 NMNPIs) and *Eunicea* (35 NMNPIs), whereas, in the Indian Ocean, the contributions of soft corals of genus *Sarcophyton* (45 NMNPIs) and sponges of the genus *Hyrtios* (24 NMNPIs) were notable, yielding the highest number of NMNPIs. In the Pacific Ocean, the top three genera were all soft corals, namely *Sinularia* (373 NMNPIs), *Sarcophyton* (NMNPIs 206), and *Lobophytum* (144 NMNPIs), totaling 16.6% of all NMNPIs recorded in this ocean.

The vast majority of NMNPIs reported during the period of 2010–2019 (68.9%) were associated with Asian countries, reinforcing the fact that these countries are the main drivers of NMNPIs discovery (accounting for 37.2% from 1990–1999 and 56.1% from 2000–2009). In comparison with the period 2000–2009, the overall ranking of continents in finding NMNPIs has remained unaltered, with Asia (69.1%) being followed by Oceania (14.3%), America (7.9%), Africa (4.6%), Europe (3.2%), and Antarctica (0.9%). For the absolute number of NMNPIs, all continents, apart from Asia, experienced a decrease in the number of NMNPIs being reported, with America decreasing by 42.6% and Antarctica by 35.0%. The Porifera yielded the majority of NMNPIs in all continents between 2010 and 2019, except for Antarctica and Asia, where the Mollusca and the Cnidaria dominated, respectively.

#### 2.2.3. Country and Exclusive Economic Zones

The top five EEZs that yielded the most NMNPIs from 2010–2019 are located in China (1390 NMNPIs), Taiwan (792 NMNPIs), Japan (409 NMNPIs), Australia (392 NMNPIs) and Indonesia (349 NMNPIs), with the Chinese EEZ alone accounting for nearly one-quarter (24.7%) of all NMNPIs recorded during this decade (Figure 5). Never has a single country played such a dominant role in the discovery of NMNPIs as China during the 2010s. Japan, which led the reporting of NMNPIs during the 1990s and the 2000s (with 21.7 and 13.5% of all NMNPIs reported during those decades, respectively), currently ranks third, reporting less than 40% of NMNPIs compared to the 2000s (717 NMNPIs) and less than 60% compared to the 1990s (977 NMNPIs). It is worth highlighting the change which has occurred in Vietnam, where there had previously been no reports of NMNPIs during the 1990s compared with 336 NMNPIs reported during 2010–2019. An interdecadal comparison of the number of NMNPIs by country is presented in Table 3.

#### 2.2.4. Biodiversity Hotspots and Large Marine Ecosystems

The Indo-Burma BH was the most dominant for the biodiscovery of NMNPIs from 2010–2019, accounting for more than 32.3% of these biomolecules (1819 NMNPIs) and largely surpassing the most dominant BH from the 1990s and the 2000s (the Sea of Japan and the Caribbean Islands, Figure 6). The proportion of reported NMNPIs originating from Sundaland and Wallacea during the 2010s was higher when contrasted to the 2000s (134 vs. 28 NMNPIs and 235 vs. 95 NMNPIs, respectively). The Sea of Japan’s relevance in this area has been dropping over the years, with the number of NMNPIs reported from this BH during the 2010s being less than half of that reported in the 1990s (407 vs. 980 NMNPIs).

The two most relevant LME for the biodiscovery of NMNPIs during the 2010s were the South China Sea (2091 NMNPIs) and the Kuroshio Current (611 NMNPIs), representing 37.1 and 10.9%, respectively, of all NMNPIs yielded during this decade. The prevalence of the South China Sea in the biodiscovery of NMNPIs was already perceptible during the 2000s when it had already surpassed the Kuroshio Current, which was the most relevant LME during the 1990s (Figure 7). The biodiscovery of NMNPIs from three of the most significant LMEs (the Caribbean Sea, the East China Sea, and the Mediterranean Sea) dropped by nearly 50% between the 2000s and the 2010s.

## 3. Discussion

From 2010–2019, marine invertebrates continued to be a valuable source of NMNP. Sponges (Porifera), soft corals and sea fans (Cnidaria) continued to yield the majority of NMNPIs reported in the past decade, confirming previous trends [9]. The record number of NMNPIs reported during the 2010s (5630) shows that much remains unknown about the chemical diversity of marine invertebrates. Given the unprecedented threats that the world’s oceans face [16], one can but wonder how much of this marine chemodiversity may be lost forever, perhaps even before being known to science. This unfortunate scenario is even more dramatic in coral reefs, which continue to endure devastating coral bleaching effects triggered by ocean warming and other anthropogenic stressors [17]. These highly biodiverse ecosystems, namely in the Indo-Pacific region, have reinforced their role as leading sources of NMNPIs, especially within the Indo-Burma BH. The China Sea is a region now yielding the highest proportion of NMNPIs, which is likely a direct consequence of China assuming the lead bioprospecting role for discovering these biomolecules during the 2010s.

After a peak number of 704 NMNPIs recorded in a single year (2012), the number of these biomolecules recorded in subsequent years has been steadily decreasing. Whether this decreasing trend has resulted from lower bioprospecting efforts, or the potential exhaustion of chemical diversity yielded from traditional marine invertebrate sources remains open to investigation. Theoretically, the chemical diversity featured by marine invertebrates is a finite variable, and the discoveries of NMNPIs will most likely plateau and subsequently start to decrease. It is certainly worth investigating what trends will emerge during the 2020s. Nonetheless, we may have to wait until half of the present decade has passed before understanding if there is indeed a decrease in the yielding of NMNPIs, as data from 2020–2022 could be biased due to the limitations imposed by the COVID-19 pandemic; a caveat already acknowledged in other research fields [18].

## 4. Materials and Methods

We have surveyed the MarinLit database [19] to expand the time frame used to survey NMNP and added new information to Leal et al. [9] spanning from 2010–2019. The following information was retrieved for each NMNPIs: source organism, taxonomic information, and collection site. Taxonomic and biogeographic information was retrieved from each publication on the MarinLit database for marine invertebrates and the World Register of Marine Species (WoRMS) database [15] was used to provide the most up-to-date taxonomic classification available. While tunicates belong to the phylum Chordata, previous works addressing NMNPIs from marine invertebrates also cover tunicates. As such, this group of marine organisms was also included in the present study, and each time phylum Chordata is mentioned in the text, it solely refers to tunicates.

Information referring to the collection site where the specimens were screened to describe NMNPIs was used to determine origins related to country, continent, ocean, latitude, exclusive economic zone (EEZ), large marine ecosystem (LME), and biodiversity hotspot (BH). A total of six continents (Africa, America, Antarctica, Asia, Europe, and Oceania) and five oceans (Antarctic, Arctic, Atlantic, Indian, and Pacific) were considered for the present work. Concerning latitude, locations of origin were identified as polar (above the Arctic Circle and below the Antarctic Circle), temperate (between the Tropic of Cancer and the Arctic Circle and between the Tropic of Capricorn and the Antarctic Circle), or tropical (between the Tropic of Cancer and the Tropic of Capricorn), as well as northern or southern hemisphere. Concerning EEZs, data of external territories (e.g., provinces, overseas departments etc.) were isolated from their parent country and treated as a separate EEZ. All geographical information reported in the present work was mapped using ArcGIS 10.8 software.

## 5. Conclusions

It is possible that interest in the bioprospecting of marine invertebrates for NMNP may have slowed down due to the growing interest in the bioprospecting of marine microorganisms (e.g., bacteria, cyanobacteria, and fungi) [2]. In general, the controlled production of the biomass of target microorganisms yielding natural products of interest is easier to scale up than that of marine invertebrates [20]. The fact that both sponges and cnidarians host remarkably diverse and chemically-bioactive communities of microorganisms [21,22,23] may also have contributed to these marine invertebrates maintaining a relevant role in the bioprospecting landscape of the marine realm. Whether the remarkable advances made by the omics toolbox [24,25,26] and artificial intelligence [27] may disrupt how we screen the marine realm for its chemical diversity in the 2020s remains an open question.

## Figures and Tables

**Figure 1 marinedrugs-20-00389-f001:**
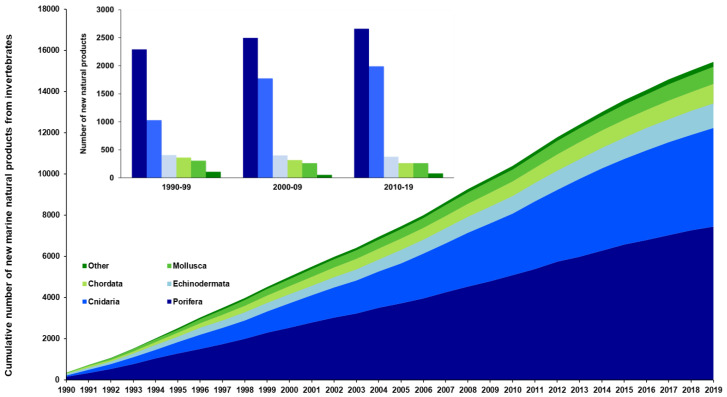
Cumulative number of new marine natural products discovered from invertebrate phyla between 1990 and 2019 (Chordata solely refers to tunicates; “Other” refers to the phyla Annelida, Bryozoa, Arthropoda, Brachiopoda, Hemichordata, and Platyhelmintes).

**Figure 2 marinedrugs-20-00389-f002:**
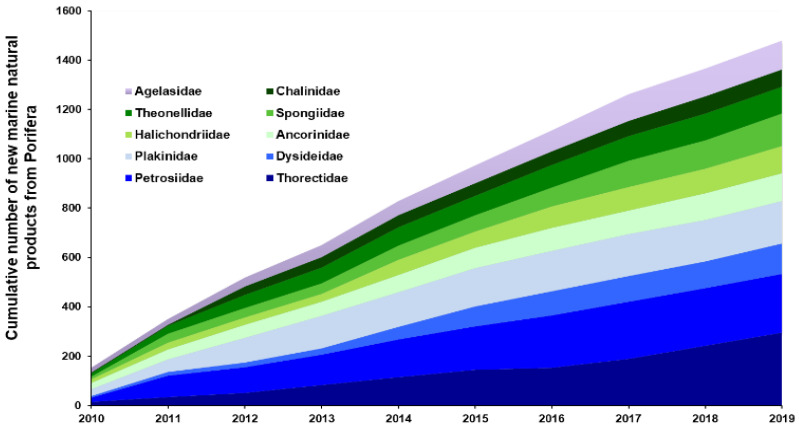
Cumulative number of new marine natural products discovered from the most representative families within the phylum Porifera (Agelasidae, Ancorinidae, Chalinidae, Dysideidae, Halichondriidae, Petrosiidae, Plakinidae, Spongiidae, Theonellidae, and Thorectidae).

**Figure 3 marinedrugs-20-00389-f003:**
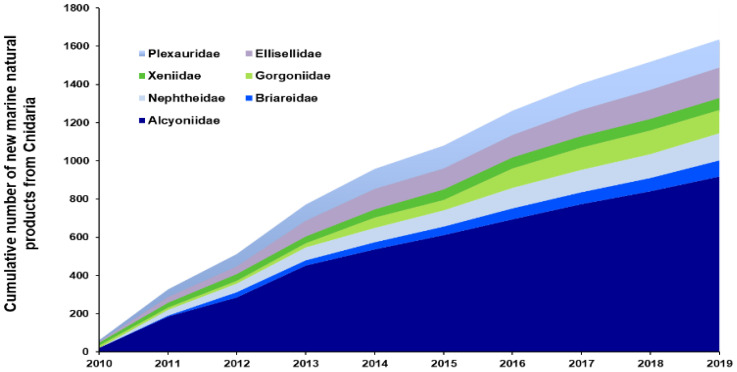
Cumulative number of new marine natural products discovered from the most representative families within the phylum Cnidaria (Alcyoniidae, Briareidae, Ellisellidae, Gorgoniidae, Nephtheidae, Plexauridae, and Xeniidae,) from 2010–2019.

**Figure 4 marinedrugs-20-00389-f004:**
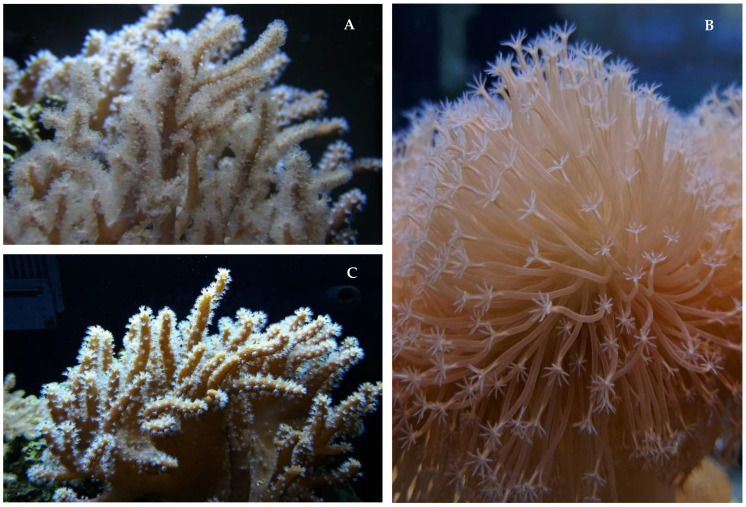
The three genera of marine invertebrates that yielded more new marine natural products from 2010–2019; soft corals within family Alcyoniidae, phylum Cnidaria: (**A**) *Sinularia*; (**B**) *Sarcophyton*; and (**C**) *Lobophyton*.

**Figure 5 marinedrugs-20-00389-f005:**
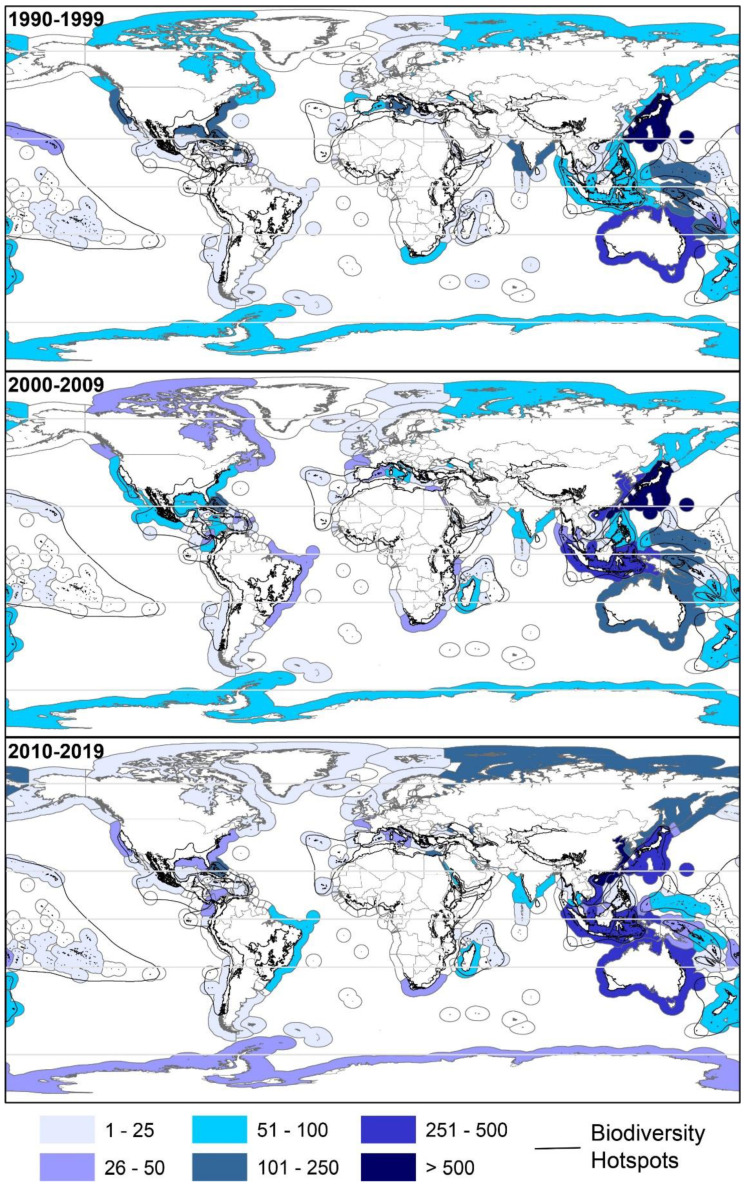
Cumulative number of new marine natural products from invertebrates discovered within different Economic Exclusive Zones during the 1990s, 2000s and 2010s. The Boundaries of biodiversity hotspots worldwide are also presented.

**Figure 6 marinedrugs-20-00389-f006:**
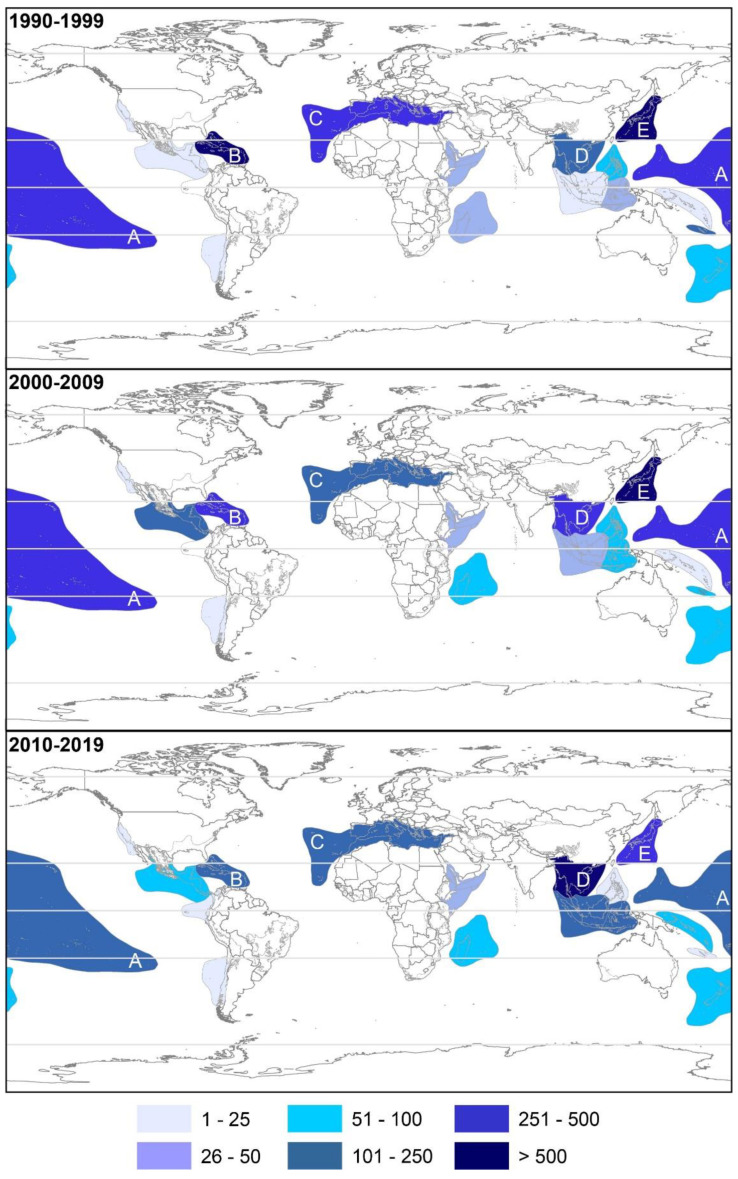
Cumulative number of new marine natural products from invertebrates discovered in different biodiversity hotspots (BH) during the 1990s, 2000s, and 2010s. (A) Polynesia-Micronesia, (B) Caribbean Islands, (C) Mediterranean Sea, (D) Indo-Burma, and (E) Sea of Japan.

**Figure 7 marinedrugs-20-00389-f007:**
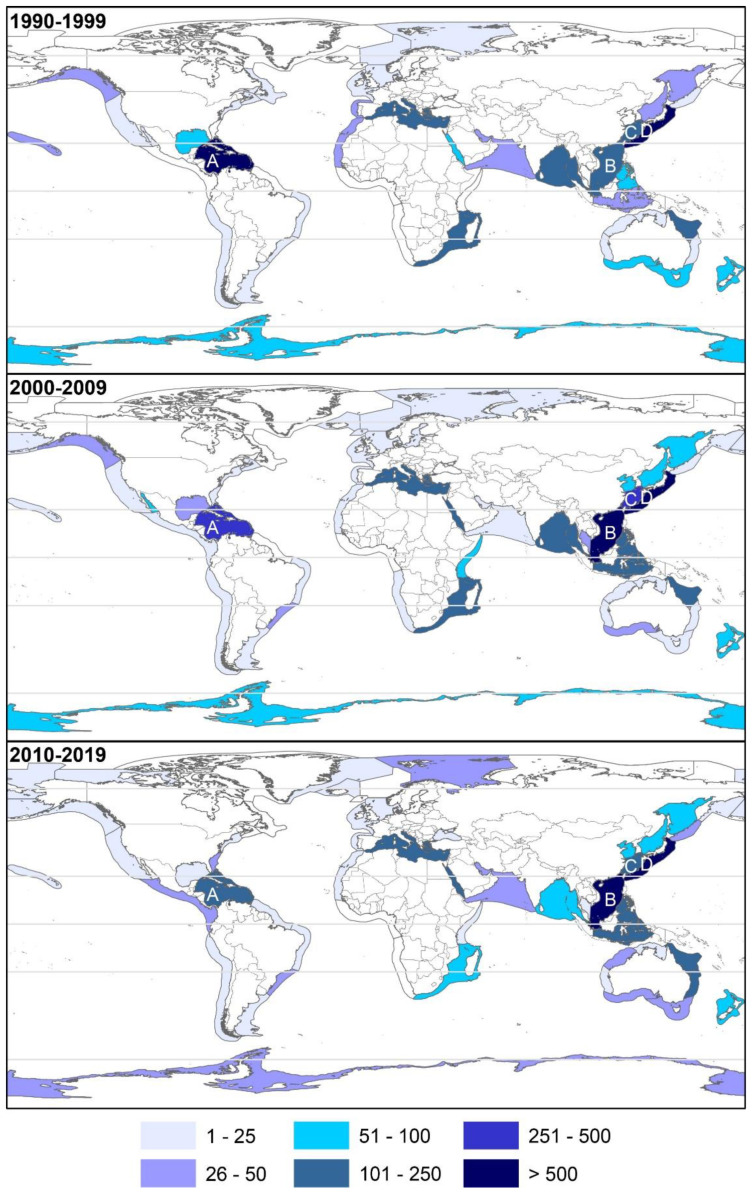
Cumulative number of new marine natural products from invertebrates discovered in different large marine ecosystems (LMEs) during the 1990s, 2000s, and 2010s. (A) Caribbean Sea, (B) South China Sea, (C) East China Sea, and (D) Kuroshio Current.

**Table 1 marinedrugs-20-00389-t001:** Number of new marine natural products (NMNP) discovered from the most representative taxa of within phyla Cnidaria and Porifera from 1990 to 2019 and their decadal variation.

Taxon	Common Name	NMNP			Decadal Variation (%)		
		1990s	2000s	2010s	1990s 2000s	1990s 2010s	2000s 2010s
Phylum Cnidaria		1031	1773	1989	+72	+93	+12
Class Anthozoa		1017	1758	1977	+73	+94	+12
Sub-class Octocorallia		963	1715	1901	+78	+97	+11
Order Alcyonacea		934	1694	1896	+81	+103	+12
Family Alcyoniidae	Soft corals	293	489	1001	+67	+242	+105
Family Ellisellidae	Sea fans	29	97	160	+234	+452	+65
Family Gorgoniidae	Sea fans	109	165	116	+51	+6	−30
Family Nephtheidae	Soft corals	58	227	160	+291	+176	−30
Family Plexauridae	Sea fans	97	99	157	+2	+62	+59
Phylum Porifera	Sponges	2291	2496	2659	+9	+16	+7
Class Demospongiae		2287	2492	2416	+9	+6	−3
Sub-class Heteroscleromorpha		1654	1755	1596	+6	−4	−9
Order Agelasida		85	56	122	−34	+44	+118
Family Agelasidae		77	55	118	−29	+53	+115
Order Haplosclerida		424	403	413	−5	−3	+2
Family Petrosiidae		209	209	239	0	+14	+14
Order Tetractinellida		NA	NA	322	NA	NA	NA
Family Ancorinidae		86	165	113	+92	+31	−32
Family Theonellidae		133	71	109	−47	−18	+54
Sub-class Keratosa		488	610	308	+25	−37	−50
Order Dictyoceratida		488	610	606	+25	+24	−1
Family Thorectidae		175	261	295	+49	+69	+13
Order Homosclerophorida		110	155	178	+41	+62	+15
Family Plakinidae		110	155	173	+41	+57	+12

**Table 2 marinedrugs-20-00389-t002:** Number of new marine natural products (NMNP) discovered per decade from the most representative taxa of marine invertebrate phyla (other than the Porifera and the Cnidaria) from 1990–2019 and their decadal variation.

Taxon	Common Name	NMNP			Decadal Variation (%)		
		1990s	2000s	2010s	1990s 2000s	1990s 2010s	2000s 2010s
Phylum Echinodermata		409	400	379	−2	−7	−5
Class Asteroidea	Sea stars	293	236	164	−19	−44	−31
Order Valvatida		106	138	115	+30	+8	−17
Family Oreasteridae		24	24	72	0	+200	+200
Class Holothuroidea	Sea cucumbers	86	127	181	+48	+110	+43
Order Dendrochirotida		55	53	129	−4	+135	+143
Family Sclerodactylidae		3	0	64	−100	+2033	NA
Family Cucumariidae		48	46	32	−4	−33	−30
Phylum Chordata		361	317	262	−12	−27	−17
Class Ascidiacea	Sea squirts	361	317	262	−12	−27	−17
Order Aplousobranchia		299	264	192	−12	−36	−27
Family Didemnidae		117	123	82	+5	−30	−33
Family Polyclinidae		49	63	53	+29	+8	−16
Phylum Mollusca		308	263	262	−15	−15	−0
Class Gastropoda		279	193	230	−31	−18	+19
Order Nudibranchia	Sea slugs	116	53	146	−54	+26	+175
Family Chromodorididae		54	11	69	−80	+28	+527
Phylum Annelida	Bristle worms	22	7	4	−68	−82	−43
Phylum Bryozoa	Moss animals	71	42	73	−41	+3	+74
Phylum Arthropoda	Crustaceans	0	4	1	-	-	-
Phylum Brachiopoda		4	0	0	-	-	-
Phylum Hemichordata		9	0	1	-	-	-
Phylum Platyhelmintes	Flatworms	2	2	0	-	-	-

**Table 3 marinedrugs-20-00389-t003:** Number of new marine natural products from invertebrates (NMNPIs) discovered in the most important exclusive economic zones (EEZs) from 1990–2019 and their decadal variation.

EEZ	NMNPIs			Decadal Variation (%)		
	1990s	2000s	2010s	1990s 2000s	1990s 2010s	2000s 2010s
Antarctica	97	80	49	−18	−49	−39
Australia	441	248	392	−44	−11	+58
Bahamas	167	125	122	−25	−27	−2
Brazil	12	46	51	+283	+325	+11
China	27	397	1390	+1370	+5048	+250
Colombia	0	87	34	NA	NA	−61
Egypt	12	41	139	+242	+1058	+239
Fiji	63	60	40	−5	−37	−33
France	16	31	44	+94	+175	+42
India	147	87	57	−41	−61	−34
Indonesia	59	286	349	+385	+492	+22
Italy	114	80	42	−30	−63	−48
Japan	977	717	409	−27	−58	−43
Madagascar	1	57	65	+5600	+6400	+14
Malaysia	3	9	36	+200	+1100	+300
Micronesia	211	105	79	−50	−63	−25
New Zealand	62	64	72	+3	+16	+13
Palau	83	80	44	−4	−47	−45
Panama	0	33	44	NA	NA	+33
Papua New Guinea	104	124	28	+19	−73	−77
Russia	55	87	123	+58	+124	+41
Saudi Arabia	7	13	55	+86	+686	+323
Solomon Islands	19	17	55	−11	+189	+224
South Africa	99	41	27	−59	−73	−34
South Korea	77	350	172	+355	+123	−51
Taiwan	54	684	792	+1167	+1367	+16
Thailand	5	38	71	+660	+1320	+87
USA	106	63	33	−41	−69	−48
Vietnam	1	22	336	+2100	+33,500	+1427

## Data Availability

All data used in the present study are available in the MarinLit database (https://marinlit.rsc.org/).

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
