# Peer review of "Updated Trends on the Biodiscovery of New Marine Natural Products from Invertebrates"

_marinedrugs, 2022, doi:10.3390/md20060389_

Round 1
Reviewer 1 Report
This manuscript describes research trends of marine natural products.
The authors analyzed the trends of new marine natural products from invertebrates over the past 30 years, and organized each data into tables and figures for easy understanding.
I think that such analysis data will be of great help to researchers who study and plan marine natural products research.
However, the 1990s and 2010s in the decadal variation of Tables 1, 2, and 3 make it rather difficult to understand. It would be better to display only 1990s 2000s, 2000s 2010s
And it is recommended that the expressions 'between 2010-2019' or 'between 2010 and 2019' be replaced with 'in 2010-2019' or 'from 2010 to 2019'.
Author Response
This manuscript describes research trends of marine natural products.
The authors analyzed the trends of new marine natural products from invertebrates over the past 30 years, and organized each data into tables and figures for easy understanding.
I think that such analysis data will be of great help to researchers who study and plan marine natural products research.
Authors reply: We sincerely thank Reviewer 1 for acknowledging the quality of our study.
However, the 1990s and 2010s in the decadal variation of Tables 1, 2, and 3 make it rather difficult to understand. It would be better to display only 1990s 2000s, 2000s 2010s
Authors reply: We acknowledge the constructive criticism by Reviewer 1 on the way we have presented decadal variation of Tables 1, 2, and 3. However, we wanted to compare trends over the three decades covered in our work, hence the comparisons 1990s 2000s, 1990s 2010s and 2000s 2010s. Moreover, as this is a follow-up of a previous study by two of the authors (R Calado and M C Leal who authored “Leal, M. C.; Puga, J.; Serodio, J.; Gomes, N. C. M.; Calado, R., Trends in the discovery of new marine natural products from invertebrates over the last two decades - Where and what are we bioprospecting? PLOS One 2012, 7, (1), e30580”) we have decided to maintain the same structure and rationale of the tables being displayed so readers can more easily refer to these two publications.
And it is recommended that the expressions 'between 2010-2019' or 'between 2010 and 2019' be replaced with 'in 2010-2019' or 'from 2010 to 2019'.
Authors reply: Corrected as suggested by Reviewer 1 throughout the manuscript.
Reviewer 2 Report
Calado et al. describe the trends on biodiscovery of new natural products produced by marine invertebrates. This paper summarizes the new natural compounds isolated from marine sources in aspect of taxonomical trends, bioprospecting efforts and natural products richness in terms of geographical trends, latitudinal, different countries and exclusive economic zones, biodiversity hotspots and large marine ecosystems. The problems and prospects of taxonomical identification of marine invertebrates also have been focused. Additionally, geographical distribution of marine sources has been discussed.
After careful reading the manuscript, I would recommend the manuscript to accept for publication in Marine Drugs with the following minor observation/suggestions:
1. It would be better to include the data of 2020
2. Please follow the unique style of references of Marine Drugs. The references are not in unique style
Author Response
Calado et al. describe the trends on biodiscovery of new natural products produced by marine invertebrates. This paper summarizes the new natural compounds isolated from marine sources in aspect of taxonomical trends, bioprospecting efforts and natural products richness in terms of geographical trends, latitudinal, different countries and exclusive economic zones, biodiversity hotspots and large marine ecosystems. The problems and prospects of taxonomical identification of marine invertebrates also have been focused. Additionally, geographical distribution of marine sources has been discussed.
After careful reading the manuscript, I would recommend the manuscript to accept for publication in Marine Drugs with the following minor observation/suggestions:
Authors reply: We sincerely thank Reviewer 2 for such positive feedback on our study.
- It would be better to include the data of 2020
Authors reply: While we acknowledge Reviewer 2 suggestion, 2020 is outside the scope of our research (the three decades: 1990s, 2000s and 2010s), as it is the first year of the 2020s and should be addressed in the follow up study covering this ongoing decade.
- Please follow the unique style of references of Marine Drugs. The references are not in unique style
Authors reply: As recommended by Reviewer 2, the style of the reference list was corrected to match the guidelines provided by Marine Drugs.